# Feasibility of low-field magnetic resonance imaging (lf-MRI) for longitudinally evaluating experimentally induced lumbar intervertebral disc injuries in goat models (*Capra hircus*): A pilot study

Jeryl C. Jones[ORCID][1☯¤a*], Mario J. Krussig[2☯¤a], Matthew W. Breed[3], Cerano D. Harrison[4¤b], John W. Gilpin[5,6], Guillermo M. Rimoldi[7], Jeremy J. Mercuri[2], Ahmed A. B. Ali[1], William C. Bridges[8]

1 Department of Animal and Veterinary Sciences, Clemson University, Clemson, South Carolina, United States of America, 2 Department of Bioengineering, Clemson University, Clemson, South Carolina, United States of America, 3 Office of Animal Resources, Division of Research, Clemson University, Clemson, South Carolina, United States of America, 4 South Carolina Translational Research Improving Musculoskeletal Health Center, Clemson University, Clemson, South Carolina, United States of America, 5 Clemson University School of Health Research, Clemson University, Clemson, South Carolina, United States of America, 6 University of South Carolina School of Medicine, Greenville, Greenville, South Carolina, United States of America, 7 Clemson Veterinary Diagnostic Center, Clemson University, Columbia, South Carolina, United States of America, 8 School of Mathematical and Statistical Sciences, Clemson University, Clemson, South Carolina, United States of America

☯ These authors contributed equally to this work.
¤a Current address: Poly-Med, Anderson, SC, United States of America
¤b Current address: Office of the Vice President for Research, University of South Carolina, Columbia, SC, United States of America
* jerylj@clemson.edu

## Abstract

Intervertebral disc injury and degeneration are among the most common causes of lower back pain and debilitation in humans. This prospective, descriptive, pilot study was designed to support our team's long-term research goals of measuring effects of novel therapies for lumbar disc injury and degeneration using small ruminant translational research models. Our overall aim was to determine whether low-field magnetic resonance imaging (lf-MRI) would be a feasible technique for longitudinally assessing surgical microdiscectomy-induced lumbar disc injury and degeneration in goat models (*Capra hircus*). Four, female, skeletally mature, Nubian-breed goats were used. One goat was used to refine and standardize imaging and surgical protocols and the remaining three were scanned one week before and 3, 6, and 12 weeks after surgery in which two discs were injured via microdiscectomy. Gross pathologic and histologic assessments for all discs were performed at the 12-week time point. We introduced a standardized lf-MRI image acquisition protocol that required 30–60 minutes (median 47.5 minutes) and yielded good quality images. We also introduced standardized protocols for quantifying disc height index (DHI) and MRI index values from lf-MRI

**Data availability statement:** All relevant data are within the paper and its Supporting information files.

**Funding:** Funding support was provided by the South Carolina Center of Biomedical Research Excellence for Translational Research Improving Musculoskeletal Health (SC TRIMH, NIH NIGMS P20 GM121342) and the South Carolina Bioengineering Center for Regeneration and Formation of Tissues (SC BioCRAFT, NIH NIGMS). This material is also based upon work supported by the NIFA/USDA, under Multistate Project number SC- 1700608 (NC-1029). Technical Contribution No. 7433 of the Clemson University Experiment Station. •NIH NIGMS = National Institutes of Health, National Institute of General Medical Sciences •NIFA/USDA = National Institute of Food and Agriculture, United States Department of Agriculture •The funders had no role in study design, data collection and analysis, decision to publish, or preparation of the manuscript.

**Competing interests:** The authors have declared that no competing interests exist.

images. All animals tolerated anesthesia well with no signs of distress. Two of the 3 goats studied longitudinally developed unexpected, non-infectious discospondylitis at the operated disc locations. The lf-MRI characteristics of non-infectious discospondylitis in goats have not been previously reported. These findings can be used as background for future studies evaluating the feasibility of using lf-MRI as a technique for longitudinally measuring IVD injury and degeneration in goat models.

## Introduction

Lumbar intervertebral disc (IVD) injury and subsequent degeneration affect over 50% of people over the age of 50 in the United States and an estimated 4% of the global population [1]. The IVD degeneration manifests initially within the central nucleus pulposus (NP) region of the IVD. During this process, the NP extracellular matrix progressively degenerates concomitant with a loss in its water content driven by detrimental phenotypic changes in local cells. Due to the important socioeconomic impact of IVD degeneration, elucidating the underpinning pathophysiologic mechanisms involved and evaluating potential intradiscal therapies employing relevant *in vivo* models is warranted.

Small ruminants (i.e., goats and sheep) are well established translational research models for studies of lumbar IVD injury and degeneration [2–8]. These models demonstrate biochemical, histological and clinical diagnostic imaging findings comparable to those observed in humans. Clinical diagnostic imaging techniques for assessing lumbar IVDs in small ruminants have previously included radiography and high-field strength magnetic resonance imaging (hf-MRI) [2,3,8]. In humans and small ruminants, qualitative imaging characteristics of IVD degeneration include disc space narrowing, decreased nucleus pulposus signal intensity, foraminal stenosis, IVD protrusion or extrusion, endplate damage, and nerve compression [2,6,7,9–15]. Quantitative imaging measures of IVD degeneration include disc height index (DHI), MRI Index, Pfirrmann classification, and Modic scoring [2,6,7,9–15]. Low field MRI offers advantages over hf-MRI such as lower purchase costs, smaller footprints, lower maintenance costs, and open designs to accommodate larger animals and facilitate anesthesia monitoring [12,16]. Two previous studies in humans demonstrated good inter-rater agreement for lf-MRI versus hf-MRI for characterizing IVDs [17,18]. Another study in dogs reported a sensitivity of 100% and specificity of 79% for lf-MRI detection of IVD degeneration, using histopathology as the gold standard [19]. At the time of this study, no published information could be found on the use of lf-MRI for assessing lumbar IVD degeneration in small ruminant models.

Our overall aim was to determine whether lf-MRI would be a feasible technique for longitudinally assessing surgical microdiscectomy-induced lumbar IVD injury and degeneration in goat models (*Capra hircus*). This aim was partially achieved; however, two goats developed unexpected and previously unreported non-infectious discospondylitis at injured disc locations that precluded longitudinal assessment of qualitative and quantitative lf-MRI characteristics of IVD degeneration.

## Materials and methods

### Selection and description of subjects

The study was a prospective, descriptive, pilot study design and part of a larger study on effects of novel treatments for IVD injury and degeneration. All procedures were approved by and conducted in accordance with requirements of an institutional animal care and use committee in an AAALAC accredited and USDA registered research facility (Clemson University, AUP2022−0342). Procedures were also in accordance with the PHS Policy on the Care and Use of Laboratory Animals (PHS Assurance Number D16-00435). All surgeries were performed under general anesthesia, and all efforts were made to minimize animals experiencing pain and distress. Four, female, skeletally mature (approximately 3 years of age), Nubian-breed goats were selected for inclusion (Palladium BioLabs) (Flowchart provided in Fig 1). Animals were considered healthy based on temperament, appetite, general stature and absence of Q fever (*Coxellia burnetti*) (ELISA, IDEXX). The vendor measured maximum diameter of the goats at the levels of the caudal ribs and pelvis to ensure that the animal was less than 11 inches to fit in the lf-MRI.

All animal care and clinical procedures were overseen by the research center's attending veterinarian (MB) and trained animal research center staff in accordance with the approved IACUC protocol. Animals were provided with daily positive interactions and monitored at least twice daily. The animals were socially housed when possible, based on compatibility. When singly housed, animals had visual, tactile and auditory access to conspecifics and additional environmental enrichment was provided. The animals were housed in specifically designed raised pens and were provided *ad libitum* hay. A commercial ration (Purina Goat Chow, Purina Inc.) was provided twice daily as well as additional enrichment treats.

### Data recording and analysis

**Acute study.** One of the four goats (ID # 78) was selected for an acute study to develop standardized lf-MRI image acquisition and surgical protocols for use in subsequent longitudinal studies. The goat was premedicated with

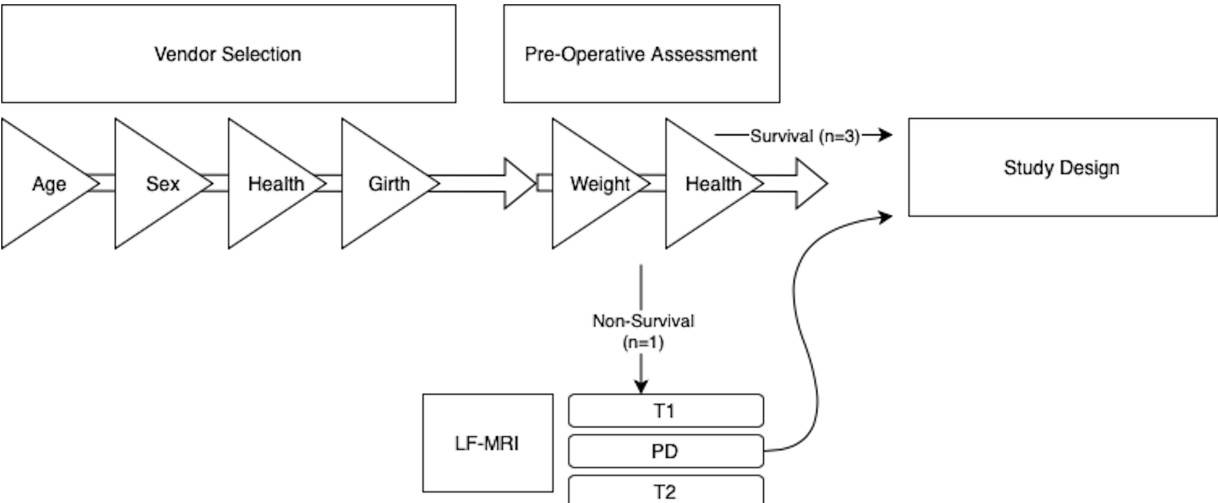

**Fig 1. Flowchart illustrating animal selection criteria.** All four Nubian goats were selected for numerous variables by both the vendor and downstream throughout the study. Age was selected to be at least 2 years of age to ensure skeletal maturity. Same sex, females, were selected to minimize variability. Following the arrival of the animals to the research facility continual weight monitoring and health assessments were conducted. Animals were considered healthy based on temperament, appetite, and general stature and absence of Q fever (*Coxellia burnetti*). The largest of the four (Goat #78) was used for a non-survival study evaluating effects of varying positioning techniques and image acquisition technical parameters on lf-MRI image quality. Protocols yielding the highest quality images were standardized and used for the three survival study goats (Goats #77, 79, 80).

acepromazine (0.075 mg/kg IM) and anesthesia was induced with diazepam (0.11 mg/kg IV) and ketamine (7.5 mg/kg IV). The goat was then intubated and maintained under anesthesia with isoflurane (2–5%), and monitored continuously. A rumen tube was placed to help minimize bloating and prevent aspiration. A lf-MRI image acquisition protocol was designed and optimized by an ACVR-certified veterinary radiologist (JJ), based on the following goals: maximize spatial resolution for lumbar IVD and vertebral endplate margin discrimination, maximize signal intensity for the normal nucleus pulposus, and minimize general anesthesia time. A detailed description of the standardized image acquisition protocol is provided in S1 Appendix. The goat was positioned in right lateral recumbency on the MRI scanner's table (Vet-MR Grande, 0.25T, open permanent magnet, Esaote, 11907 Exit 5 Parkway, Fishers, IN 46037). Positioning was then adjusted as needed to ensure the last pair of ribs and the L1-5 vertebrae were included in the scan field of view and that the vertebrae were within the scanner's isocenter for transverse, sagittal, and dorsal planes. Tuberculin syringes filled with distilled water were taped dorsal to the L1-3 vertebrae for use as water signal intensity calibration phantoms. Trial scans were acquired with varying pulse sequences and technique settings and the veterinary radiologist made initial image quality assessments. Image data were archived and the optimal image acquisition protocol was selected based on a consensus of the veterinary radiologist and other members of the research team. The protocol was then stored in the scanner's computer for use in subsequent scans.

The goat was euthanized as per the IACUC protocol immediately after completion of lf-MRI scanning. A veterinarian with 14 years of experience (AA) and a biomedical engineer with two years of experience (MK) applied sterile surgical procedures as per the IACUC protocol and refined microdiscectomy techniques described in previous publications [3,5,8]. The goat was placed in right lateral recumbency and the locations of the target lumbar IVDs were based on palpation of the last rib and lumbar spinous processes. A left ventrolateral retroperitoneal microdiscectomy was performed at each target disc using a 6 cm skin incision parallel to the lumbar vertebral transverse processes. The incision was approximately 5 cm ventral to the transverse processes and 8 cm cranial to the craniodorsal margin of the ilium (*Os Coxae*). Blunt dissection of the subcutaneous tissue was performed until the external and internal abdominal oblique muscles were visualized. Guarded dissection through the layers of the abdominal muscles followed the muscle fibers and dorsal aponeurosis to minimize bleeding and operational tears. The dorsal retroperitoneal lumbar vertebral fascia was dissected until reaching the lumbar vertebrae cranial and caudal to the target intervertebral discs. The intervertebral discs were isolated from blood vessels and nerves to avoid damage. Following intervertebral disc exposure, an 11-blade scalpel was used to create a 4 mm incision in the annulus fibrosus parallel to the endplates. A 2 mm rongeur was utilized to remove 30 mg of nucleus pulposus (approximately four sets of retrieval per disc level. The surgical site was closed using 2−0–0 suture for the muscle layer and 0 suture for subcutaneous and skin layers.

**Longitudinal study.** Three goats (ID # 77, 79, 80) were fasted the night before anesthesia, and premedicated with acepromazine (0.075 mg/kg IM). Anesthesia was induced with diazepam (0.11–22 mg/kg IV) and ketamine (7.5 mg/kg IV). Goats were intubated and maintained under anesthesia with isoflurane (2–5%), and monitored continuously. The animals were then imaged using the standardized lf-MRI image acquisition protocol developed in the acute study. Scans were acquired for each goat at each of the following time points (n = 12 scans): 1 week before surgery (baseline) and 3, 6, 12 weeks after surgery. Images for each scan were initially interpreted by the ACVR-certified veterinary radiologist (JJ) during and immediately following image acquisition. Image data were then archived for further analyses (AVS Image Analysis Laboratory, Clemson University).

Standard post-surgery analgesia was provided to all the animals using buprenorphine (0.005–0.01 mg/kg IM) every 6–8 hours and flunixin (1–2 mg/kg IM or SC) daily for up to three days post-surgery. Peri and post-operative antimicrobials were provided (ceftiofur sodium, 1.1–2.2 mg/kg IM) every 24 hours up to 3 days post-surgery. Additional analgesia (Flunixin, 1.1–2.2 mg/kg IM, or Meloxicam, 1 mg/kg PO) and antibiotics (Ceftiofur sodium, 1.1–2.2 mg/kg IM) were provided as needed at the direction of the research center's attending veterinarian (MB) during the extended follow up period. Surgical approaches and microdiscectomy procedures were performed by the same veterinarian (AA) and biomedical engineer

(MK), using the same standardized protocol developed in the acute study. The microdiscectomy procedure was performed on discs L1/L2 and L2/L3 for goats 77 and 79 and discs L2/L3 and L3/L4 for goat 80. Local anesthesia (bupivacaine) was applied to the incision upon closing at the muscle and subcutaneous layers. Animals were continually monitored until they had fully recovered (standing and walking unassisted) and were then assessed by trained research center staff twice daily.

Qualitative analyses of lf-MRI scans for each of the three goats and each of the four time points (n = 12) were performed by two observers, an ACVR-certified veterinary radiologist (JJ) and an ACR-certified medical diagnostic radiologist (JWG, Fellow of the American College of Radiology). Each observer had 30 + years of experience interpreting MRI studies. Interpretations were performed using a dedicated image analysis workstation and open-source image analysis software (Mac OS High Sierra, 10.13.6, MacPro Quad Core, Apple, Inc. Cupertino, Ca; Horos https://horosproject.org/). The order of scans was randomized, and observers were unaware of anatomic locations for IVD microdiscectomy injuries at the time of interpretation. The following qualitative findings were recorded as present or absent for each scan and each lumbar disc (L1-5), based on independent assessments: foraminal stenosis (IVD, facet hypertrophy, osteophyte), IVD protrusion or extrusion, endplate damage (superior defect, inferior defect, compression, Schmorl's Node), nerve compression (mild, moderate, or severe), and paraspinal muscle damage (mild, moderate, moderately severe, or severe). The two observers then met to resolve discordant opinions and consensus findings were used for subsequent descriptive analyses. In addition, Pfirrmann and Modic scores for each lumbar disc were recorded by the medical diagnostic radiologist (JG), when applicable.

Quantitative analyses of lf-MRI scans were performed by two observers, a laboratory manager with 6 years of image analysis experience (CH, Observer 1) and a biomedical engineer with 1 year of image analysis experience (MK, Observer 2). A standardized image analysis protocol was designed in consultation with the veterinary radiologist (JJ) and used for all measurements (see detailed, step by step descriptions with illustrations in S2 Appendix). Briefly, each observer randomized the order of scans and independently recorded the following values for each of the lumbar discs in triplicate: disc height index (DHI) [9] and MRI Index (measure of nucleus pulposus hydration) [2,7]. Observers were unaware of operated disc locations at the time of data recording. Lumbar disc locations were first identified by finding the last pair of ribs in the dorsal planar, proton-density weighted images. This location was identified the last thoracic vertebra and lumbar vertebrae were then numbered based on this landmark. Using mid-sagittal, proton-density weighted images, the pencil tool was used to hand-draw regions of interest (ROI) around the outer margins of the nucleus pulposus. The area and signal intensity values generated by the image analysis software for each ROI were then multiplied to generate the MRI index values for each IVD. The nucleus pulposus ROIs were deleted and the software's line tool was used to measure craniocaudal vertebral height at the mid-portion of each vertebra. Disc height was then measured at the dorsal, middle, and ventral locations for each disc. Two techniques were used for measuring disc height. Technique 1 was termed "endplate to endplate". For this technique, the electronic cursors were placed on the adjacent cartilaginous margins of vertebral endplates. Technique 2 was termed "bone to bone". For this technique, the electronic cursors were placed on the adjacent subchondral bone margins of vertebral endplates. This technique was intended to mimic the margins that would be used for measuring disc heights from radiographs. The DHI values were calculated using an average of the three length measurements acquired at the mid-disc, dorsal disc, and ventral disc locations and then dividing this average by the vertebral height cranial to the disc of interest.

After twelve weeks post-surgery, animals were humanely euthanized as described above, and the lumbar regions of the spine were removed *en bloc*. Following removal of the fascia, each of the lumbar IVDs were removed to include the vertebral body and endplate sections cranial and caudal to the IVD. These functional spine segments were labeled by animal ID and disc location and fixed in buffered formalin. Digital radiographs of formalin-fixed specimens were acquired by animal research center personnel. Radiographs and specimens were then submitted to a veterinary diagnostic center (Clemson Livestock Poultry Health, Columbia, SC) for pathologic assessments. An ACVP-certified veterinary pathologist

(GR) with 25 years of experience performed decalcification of samples in a formic acid containing commercial solution (Cal-Ex ® II – Fixative Decalcifier, Fisher Chemical). Following decalcification, all lumbar disc samples for each goat were sectioned sagittally through the midline to obtain slides including the intervertebral discs or samples of tissue replacing them. Locations for sections were chosen based on review of the gross specimen photographs and digital radiographs. Samples were routinely embedded in paraffin to obtain 5 micrometers thick sections that were stained with hematoxylin and eosin (H&E) (Epredia Revos Workflow – enhancing tissue processor). Images were then digitized for further analyses. The veterinary pathologist (GR), a biomedical engineer (JM) with 9 years of experience, and a biomedical engineer (MK) with 2 years of experience reviewed digital images and reached a consensus on descriptions of pathologic findings.

## Statistics

Descriptive analyses were performed by a biomedical engineer (MK) and image analysis laboratory manager (CH), in consultation with a statistician (WB, Alumni Professor of Mathematical and Statistical Sciences). Frequencies of qualitative clinical and lf-MRI findings, and medians and ranges for Pfirrmann scores, MRI index, and DHI measurements were calculated. The intra- and inter-observer repeatability for MRI Index and DHI measurements were assessed with two approaches.

The first approach was to calculate the coefficients of variation (CVs) for each observer's triplicate measurements and then compare the mean CVs between observers. To compare the means, statistical models for the CVs were developed that included a term for observer effect, and also terms for animal and disc location effects (since the observers were looking at the same animals and the same disc locations, resulting in a blocked design). The models were analyzed using analysis of variance (ANOVA) with F-tests to determine if the observer effect was statistically significant. Note the F-tests for observer were essentially "paired t-tests" for observer.

The second approach was to calculate the Intraclass Correlation Coefficients (ICC) for each observer, and an overall ICC. To calculate the ICC for each observer, statistical models for the MRI Index and DHI measurements were developed for each observer that included terms for animal and disc location effects. The models were analyzed using ANOVA to estimate the variance among groups ($V_A$) (which was variance due to animal and disc location effects) and to estimate the total variance ($V_T$) (which was variance due to animal, disc location, and triplicate measurement effects). The ICC's were estimated as the ratio $V_A/V_T$.

To calculate the overall ICC, the same process was repeated except that the observer effect was included in the model. The residuals of the models were assessed for ANOVA assumptions of the particular analysis. If the residuals were found to not satisfy the ANOVA assumptions, the analyses were repeated with rank-transformed data. As residuals met the required ANOVA assumptions, and the results based on rank-transformed data were consistent with results based on the original data, results and discussions are based on analyses using the original data.

For all F-tests, p-values less than 0.05 were considered evidence of statistical significance. All statistical calculations were performed with Microsoft® Excel, version 16.89.1, GraphPad® Prism Software, version 10.3.1, and JMP® Student Edition, version 18.2.0.

## Results

### Acute study

The lf-MRI T2-weighted and STIR images were considered to have poor image quality due to excessive noise artifacts and variable NP signal intensity. These problems could not be resolved by revising technical parameters. The T1-weighted images were considered to have good image quality and contribute helpful information for morphologic assessments, however the information gained did not outweigh the increased risk to the animal of lengthy anesthesia time. Sagittal, dorsal planar, and transverse proton-density weighted pulse sequences with customized settings recommended by the applications specialist were considered to best meet the stated goals of maximizing NP signal intensity, maximizing spatial resolution of discs and vertebrae, and minimizing anesthesia time (see S1 Appendix). Scan times for the longitudinal study

lf-MRI scans using these pulse sequences ranged from 30 to 60 minutes (median 47.5 minutes), total anesthesia times (induction to extubation) for survival MRI ranged from 85 minutes to 150 minutes (median 100 minutes).

## Longitudinal study

**Initial lf-MRI interpretations and clinical examination findings.** All animals underwent four lf-MRIs (baseline, 3-, 6- and 12-weeks post microdiscectomy) and the microdiscectomy-procedure as described above. All goats recovered well post anesthesia. Prior to surgery and baseline MRI, Goat 77 was treated for mild clinical signs of pneumonia with antimicrobials (ceftiofur sodium (1.1–2.2 mg/kg) IM). Additionally, a small swelling was noted at the incision of Goat 77 approximately 5 days post-surgery, and an additional 3 days of antimicrobials (ceftiofur sodium, 1.1–2.2 mg/kg IM every 24 hours) was provided.

Detailed descriptions of initial MRI interpretations by the veterinary radiologist for each goat and each timepoint are provided in the S3 Appendix. Example images are provided in Fig 2. For Goat 79, findings included mild characteristics consistent with expected post-operative soft tissue injury and healing and a subjectively decreased disc height and disc signal for one of the injured discs at 6- and 12-week timepoints. These findings were interpreted to

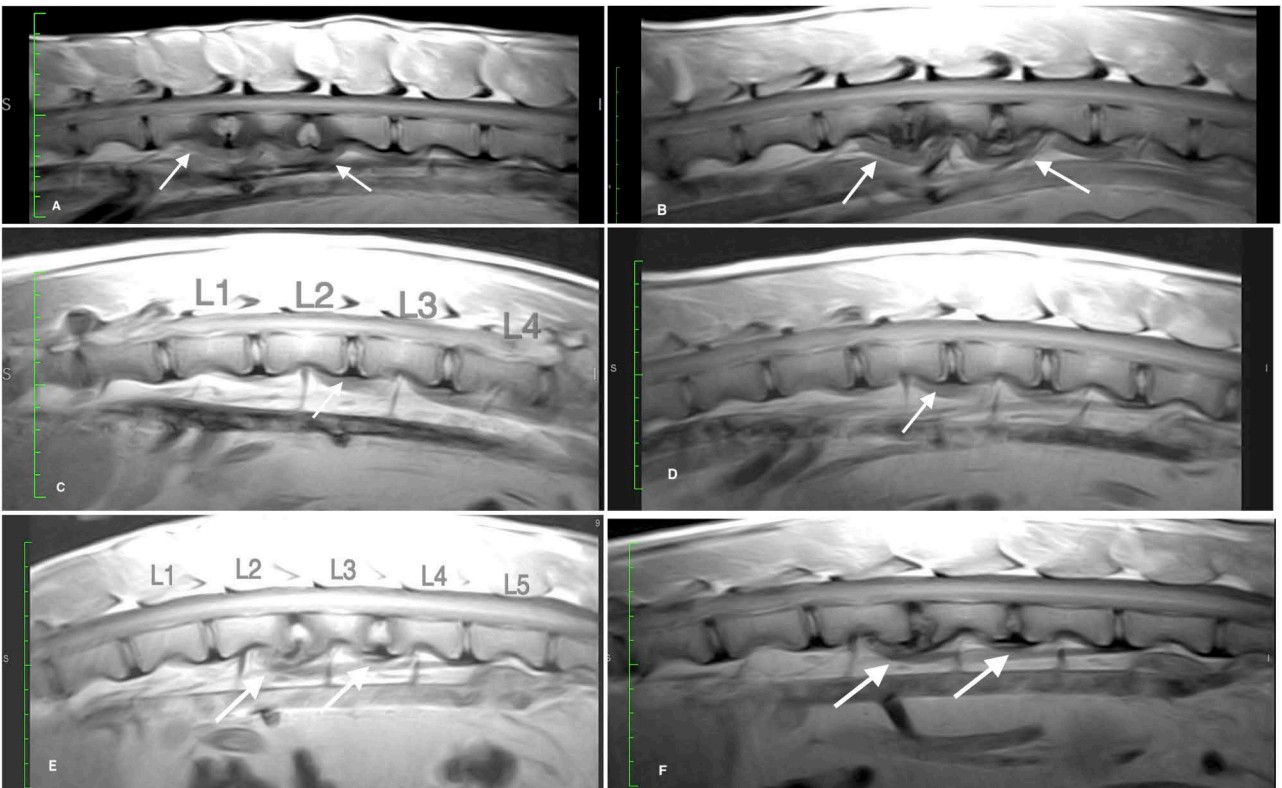

**Fig 2. Sagittal planar, proton-density weighted, lf-MRI images of the lumbar spine for each of the three goats in the longitudinal study. A.** Goat #77, 6 weeks post-operative: increased disc signal, vertebral sclerosis, annulus protrusion, and endplate lysis at L1-2 and L2-3 (arrows). **B.** Goat #77, 12 weeks post-operative: partial resolution of previous increased disc signal with progressive vertebral sclerosis, annulus protrusion, and proliferative, paravertebral tissue at L1-2 and L2-3 (arrows). **C.** Goat #79, 6 weeks post-operative: subjective decrease in disc signal L2-3 (arrow). **D.** Goat #79, 12 weeks post-operative: subjective decrease in disc signal and disc space width L2-3 (arrows). **E.** Goat #80, 6 weeks post-operative: increased disc signal, increased disc space width, loss of endplate margins, and decreased vertebral body signal at L2-3 and L3-4 (arrows). Goat #80, 12 weeks post-operative: mixed increased/decreased disc signal, increased disc space width, loss of endplate margins, decreased vertebral body signal at L2-3 and L3-4 (arrows).

be consistent with early IVD degeneration. This goat was normal clinically prior to baseline MRI and recovered well from all MRI procedures and surgery. At no time during post-surgery until the terminal MRI did Goat 79 appear painful to deep palpation or in postural adjustments, whether on or off analgesic and antimicrobial treatment. For Goats 77 and 80 at the 3-week and 6-week post-operative MRI timepoints, findings at two IVD locations included increased nucleus pulposus signal intensity, widened intervertebral disc spaces, loss of endplate margin discrimination, and increased signal intensity in adjacent paraspinal muscles. These imaging findings were interpreted to be consistent with discospondylitis.

Based on concerns for possible discospondylitis in goats 77 and 80, more frequent physical examinations were performed. Antimicrobial (Ceftiofur sodium, 1.1–2.2 mg/kg IM) and analgesic (Flunixin, 1.1–2.2 mg/kg IM or Meloxicam, 1 mg/kg PO) treatments were also provided. At the 6-week post-operative timepoint for goats 77 and 80, lf-MRI findings indicated possible progression of the discospondylitis. Based on these concerns, antimicrobials were again administered and continued for 10 days in Goat 77 and 5 days for Goat 80. Analgesics were also continued every day to every other day (Flunixin,1.1–2.2 mg/kg IM, or Meloxicam, 1 mg/kg PO) until euthanasia. At the 12-week timepoint, lf-MRI findings for the two affected goats indicated partial resolution of the previous characteristics of discospondylitis. At no time during the post-surgical period until the terminal MRI did any of the goats appear painful to deep palpation or in postural adjustments, whether on or off analgesic and antimicrobial treatment.

### Qualitative lf-MRI analyses

Results of consensus qualitative lf-MRI assessments are summarized in Table 1. Both radiologists considered the modified proton-density weighted lf-MRI pulse sequence images to be of good quality. The nucleus pulposus exhibited an increased signal intensity relative to the annulus fibrosus at all disc locations in the baseline scans. Motion artifacts were noted in 3/12 scans (25%). Consensus opinions for the anatomic locations of normal IVDs and microdiscectomy-injured IVDs were correct for 9/9 scans (100%). For Goats 77 and 80 at 3-, 6- and 12-week time points, both radiologists agreed that characteristics were consistent with discospondylitis in animals or spondylodiscitis in humans. Both radiologists considered an infectious process to be the top differential diagnosis, however a non-infectious inflammatory response was also considered. Both radiologists considered the inflammatory response characteristics to be partially resolved at the 12-week timepoint. Pfirrmann and Modic scoring were not possible for these affected discs. Additional findings identified during the consensus reading session included foraminal stenosis at some locations due to facet hypertrophy and disc margin protrusion.

**Quantitative analyses.** Results of MRI Index measurements are detailed in the S6 Appendix, summarized in Tables 2 and 3, and graphically illustrated in Figs 3 and 4. The injured disc locations for Goats 77 and 80 were excluded from measurements because non-infectious discospondylitis caused a loss of visualization of nucleus pulposus margins. Post-operative scan measurements were excluded from repeatability analyses due to a communication breakdown resulting in observers measuring different discs. Intra-observer analyses of measurements for pre-operative (Time 0) scans indicated that percentages of repeatable triplicate measurements were 58.33% for Observer 1 and 33.33% for Observer 2 (measurements were defined as repeatable if mean coefficients of variation were 10 or below). ICC analysis showed moderately high to high correlations between observers at all time points (0.63–0.86). For the inter-observer comparisons, mean CV values of the MRI index for Observers 1 and 2 were 12.339 and 11.939, respectively. Coefficients of variation were not statistically different as indicated by the p value of 0.89. While CV values surpassed values to conclude acceptable repeatability, the ICC analysis showed that triplicate measures for both observers had moderate to high correlations at all time points (obs 1: 0.73–0.82; obs 2: 0.66–0.95).

Results of Disc Height Index (DHI) measurements are detailed in S4–S6 Appendices, summarized in Tables 3–5, and graphically illustrated in Figs 5 and 6. Injured discs for Goats 77 and 80 were excluded from measurements because the

**Table 1. Results of qualitative lf-MRI analyses for each goat based on a consensus of two radiologists.**

| Goat ID # | | | 77 | | 79 | | 80 | |
|---|---|---|---|---|---|---|---|---|
| **Overall** | Total Disc Levels Evaluated | | 19 | | 20 | | 12 | |
| | Motion Artifacts | Time Points (#) | 1/ 4 | | 1/ 4 | | 2/ 4 | |
| | | Disc Levels Effected (#) | 1/ 19 | | 5/ 20 | | 3/ 12 | |
| | Pre-Surgical Notes | | Motion Artifact Noted | | Motion Artifact Noted | | L4/L5 out of Field of View | |
| **Grading** | **Uninjured Levels** | Levels Evaluated post-Surgery | 3 | T13/L1, L3/L4, L4/L5 | 3 | T13/L1, L3/L4, L4/L5 | 2 | L1/L2, L4/L5 |
| | | Foraminal Stenosis | Absent – Moderate | | Absent – Mild | | Absent – Mild | |
| | | Endplate Damage | Absent | | Absent | | Absent | |
| | | Nerve Compression | Absent | | Absent | | Absent | |
| | | Paraspinal Muscle Abnormalities | Absent | | Absent | | Absent | |
| | | Pfirrmann Grade* | 1 | | 1 | | 1 - 2 | |
| | **Injured Levels** | Levels Evaluated post-Surgery | 2 | L1/L2, L2/L3 | 2 | L1/L2, L2/L3 | 2 | L2/L3, L3/L4 |
| | | Foraminal Stenosis | Not Discernable – Mild | | Absent – Mild | | Absent – Mild | |
| | | Endplate Damage | Mild – Moderately Severe | | Absent – Mild | | Mild – Moderately Severe | |
| | | Nerve Compression | Absent – Mild – Severe | | Absent | | Absent | |
| | | Paraspinal Muscle Abnormalities | Mild – Severe | | Absent | | Mild – Moderately Severe | |
| | | Pfirrmann Grade* | Not Discernable | | 1-2 | | Not Discernable | |

Observers included one ACVR-certified veterinary radiologist and one ABR-certified medical diagnostic imaging radiologist. Observers interpreted images independently, without knowledge of the injured disc locations at the time of interpretation. The order of scans was randomized for each reading session. Observers met to resolve discordant opinions and reach a consensus.

**Table 2. Average of triplicate MRI Index measurements recorded by two observers for each goat, timepoint, and intervertebral disc level.**

| Goat ID # | Disc Level | MRI Index (Mean±SEM) | | | | | | | |
|---|---|---|---|---|---|---|---|---|---|
| | | **Baseline** | | **3 Week** | | **6 Week** | | **12 Week** | |
| | | Observer 1 | Observer 2 | Observer 1 | Observer 2 | Observer 1 | Observer 2 | Observer 1 | Observer 2 |
| 77 | L1/L2* | 28±3.9 | 25.8±1.2 | 15.5±5.5 | – | 18.2±1.8 | – | – | – |
| | L2/L3* | 26.3±1.9 | 20.9±1.8 | 17.4±2.6 | – | – | – | 32.8±5.9 | – |
| | L3/L4 | 21.2±1.9 | 16.2±1.4 | 13.3±2.1 | 15.8±0.7 | 16.0±1.3 | 13.2±2.5 | 20.4±0.9 | 18.4±2.5 |
| | L4/L5 | 16.5±2.3 | – | – | – | 17.5±1.5 | – | 16.3±2.0 | – |
| 79 | L1/L2* | 22.8±3.0 | 16.6±4.5 | 23.3±3.7 | 27.6±0.8 | 24.8±1.9 | 29.7±3.0 | 15.0±2.2 | 15.6±1.8 |
| | L2/L3* | 27.2±3.0 | 23.1±1.4 | 25.6±3.9 | 28.5±1.9 | 23.8±3.6 | 25.4±0.7 | 23.8±4.5 | 21.1±1.3 |
| | L3/L4 | 24.2±0.6 | 24.1±5.3 | 26.7±2.9 | 27.8±0.7 | 25.6±3.6 | 25.8±1.2 | 25.4±6.1 | 27.3±2.1 |
| | L4/L5 | 22.3±0.7 | 19.8±4.9 | 20.8±1.2 | 17.4±2.0 | 13.3±1.4 | – | 12.2±0.7 | 19.8±0.8 |
| 80 | L1/L2 | 19.8±4.1 | 19.8±3.5 | 14.7±0.7 | 13.9±2.0 | 16.4±1.4 | 15.8±0.7 | 15.7±1.0 | 18.6±1.3 |
| | L2/L3* | 17.0±1.0 | 24.5±0.1 | 16.7±2.3 | – | 33.3±5.3 | – | 20.7±0.9 | – |
| | L3/L4* | 15.4±3.6 | 19.1±1.6 | 16.8±3.1 | – | 42.3±11.5 | – | 30.6±1.5 | – |
| | L4/L5 | 11.4±2.3 | 16.1±1.7 | 11.1±0.6 | 10.3±1.8 | 13.7±1.8 | 13.9±1.7 | 22.6±0.7 | 12.6±0.6 |

MRI Index values are measures of nucleus pulposus hydration. Values were calculated using the following formula: maximum mid-sagittal nucleus pulposus area (mm$^2$) X mean nucleus pulposus signal intensity. Blank cells indicate locations where the observer could not sufficiently visualize nucleus pulposus margins for region of interest tracing. Asterisks indicate disc levels where experimentally induced injuries were created after baseline MRI scans.

**Table 3. Intraclass Correlation Coefficients (ICC) for triplicate bone-to-bone and endplate-to-endplate measurements performed by two observers (obs) at 4 different time points.**

| Time point | ICC (DHI endplate-endplate) | | | ICC (DHI bone-bone) | | | ICC (hydration) | | |
|---|---|---|---|---|---|---|---|---|---|
| | Between observers | Within observers | | Between observers | Within observers | | Between observers | Within observers | |
| | obs 1 & 2 | obs 1 | obs 2 | obs 1 & 2 | obs 1 | obs 2 | obs 1 & 2 | obs 1 | obs 2 |
| Baseline | 0.13 | 0.72 | 0.00 | 0.39 | 0.88 | 0.23 | 0.63 | 0.78 | 0.66 |
| Week 3 | 0.00 | 0.16 | 0.15 | 0.72 | 0.93 | 0.02 | 0.80 | 0.73 | 0.95 |
| Week 6 | 0.27 | 0.37 | 0.36 | 0.14 | 0.71 | 0.41 | 0.86 | 0.79 | 0.92 |
| Week 12 | 0.23 | 0.32 | 0.60 | 0.59 | 0.92 | 0.67 | 0.75 | 0.82 | 0.87 |

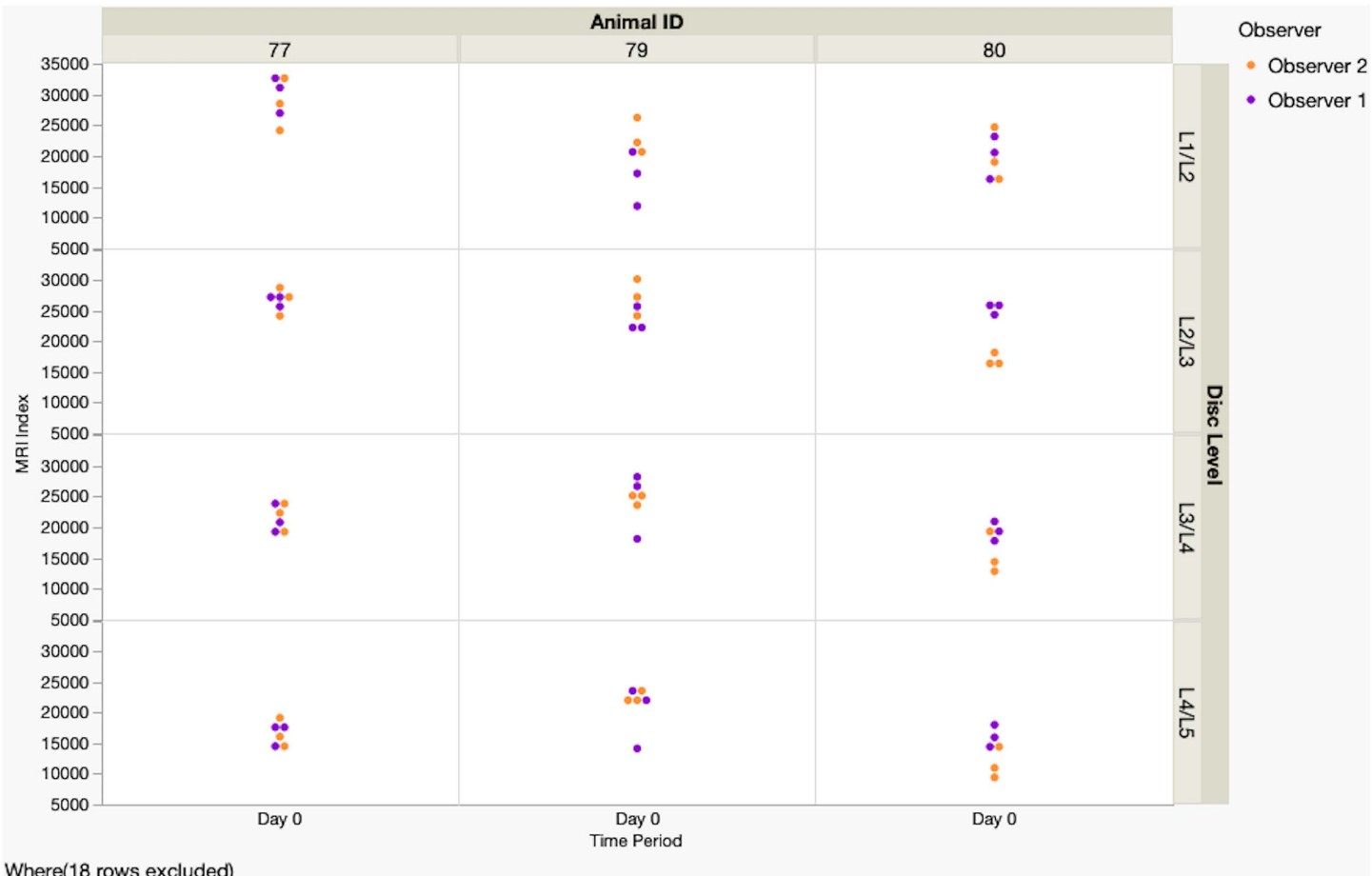

**Fig 3. Graph illustrating MRI index measurements (nucleus pulposus hydration) recorded by two observers for each intervertebral disc (L1–L5) in each of the three goats at week 0 (pre-operative).** Measurements appeared consistent for most locations; however, measurements were subjectively more variable for the L1/L2 IVD.

severe inflammatory response caused a loss of visualization of vertebral endplate and subchondral bone margins. The intra-observer repeatability analysis showed highly variable percentages of repeatable CV measures for both bone-to-bone and endplate-to-endplate measures for Observer 1 (0% − 100%). Measures for Observer 2 yielded moderate to high

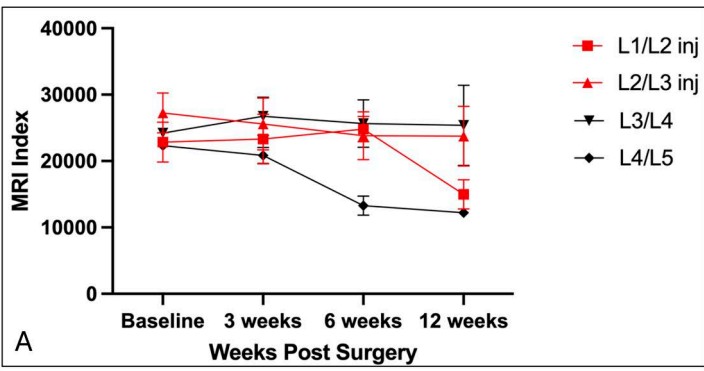

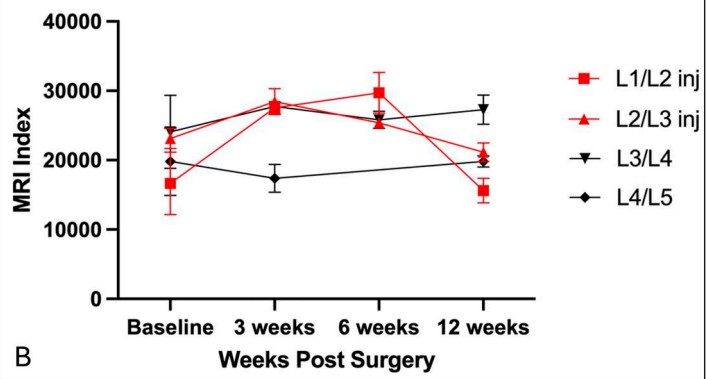

**Fig 4. Graphs illustrating MRI Index measurements at each intervertebral disc location (L1-5) and each timepoint in Goat #79.** (Goats #77 and 80 were excluded from these analyses due to non-infectious discospondylitis at injured levels causing loss of nucleus pulposus margins.). **A.** Observer 1. **B.** Observer 2. Recorded values varied between observers.

percentages of repeatable CV measures for bone-to-bone and endplate-to-endplate measures (75% − 100%). Percentages of repeatable triplicate measures were higher for Observer 2, except at time points 3 (week 6) and 4 (week 12), in that Observer 1 yielded a higher percentage of repeatability for week 6 (57.14% vs. 85.71%). Both observers had equally high percentages of repeatability for endplate-to-endplate measures at week 12 (100%). Further assessment of the intraobserver repeatability using ICC showed that, for endplate-to-endplate and bone-to-bone measurements, with the exception of endplate-to-endplate measures for week 12, Observer 1 demonstrated higher correlations of association across all four time points (ranging from .16 to .72 (e-e) and .71 to .93 (b-b)) compared to Observer 2 (ranging from 0.0 to .36 (e-e) and.02 to.67(b-b)). For the inter-observer repeatability analysis, across all time points, Observer 2 consistently demonstrated lower CV values than Observer 1 for both bone-to-bone and endplate-to-endplate DHI measurements. Statistically significant differences between observ-ers were identified for bone-to-bone measurements at day 0, week 3, and week 6, as indicated by the p values of 0.0038, 0.0188, and 0.0300, respectively. Significant differences for endplate-to-endplate measurements were only observed at day 0 (p<0.0001). At the 3- and 6-week timepoints, mean CVs for the two observers did not differ using the endplate-to-endplate measurement method. Mean CVs for the two observers did not differ using either measurement method at the 12-week timepoint (p>0.05). ICC analysis showed low to high (0.0 −.72) correlations for both bone-to-bone and endplate-to-endplate measurements. Bone-to-bone measures yielded higher ICC values than endplate-to-endplate measures.

**Pathologic assessments.** For Goat 79, no lesions were found at uninjured IVD locations and only little to no fibrous scar tissue was observed at the surgically injured locations (Fig 7). For Goats 77 and 80, characteristics of an inflammatory response with bone remodeling and dense fibrous reactive connective tissue surrounding the adjacent

**Table 4. Average of triplicate disc height index (DHI) values recorded by two observers using the bone-to-bone measurement method for each goat, time point, and intervertebral disc level.**

| Goat ID # | Disc Level | Disc Height Index Measured Bone-Bone (Mean±SEM) | | | | | | | |
|---|---|---|---|---|---|---|---|---|---|
| | | Baseline | | 3 Week | | 6 Week | | 12 Week | |
| | | Observer 1 | Observer 2 | Observer 1 | Observer 2 | Observer 1 | Observer 2 | Observer 1 | Observer 2 |
| 77 | L1/L2* | 0.305±0.006 | 0.301±0.054 | – | – | – | – | – | – |
| | L2/L3* | 0.304±0.021 | 0.293±0.092 | – | – | – | – | – | – |
| | L3/L4 | 0.285±0.005 | 0.289±0.031 | 0.310±0.014 | 0.301±0.040 | 0.305±0.017 | 0.354±0.024 | 0.286±0.032 | 0.301±0.018 |
| | L4/L5 | 0.198±0.011 | 0.326±0.107 | – | – | 0.292±0.008 | 0.290±0.014 | 0.188±0.011 | 0.240±0.012 |
| 79 | L1/L2* | 0.355±0.011 | 0.364±0.004 | 0.335±0.011 | 0.328±0.016 | 0.333±0.013 | 0.313±0.032 | 0.328±0.028 | 0.345±0.015 |
| | L2/L3* | 0.372±0.006 | 0.358±0.032 | 0.334±0.005 | 0.322±0.025 | 0.348±0.007 | 0.298±0.036 | 0.354±0.007 | 0.315±0.020 |
| | L3/L4 | 0.384±0.031 | 0.340±0.041 | 0.368±0.022 | 0.342±0.019 | 0.362±0.003 | 0.293±0.033 | 0.392±0.014 | 0.293±0.027 |
| | L4/L5 | 0.370±0.003 | 0.359±0.023 | 0.366±0.014 | 0.325±0.028 | 0.285±0.029 | – | 0.360±0.002 | 0.286±0.033 |
| 80 | L1/L2 | 0.303±0.013 | 0.360±0.044 | 0.316±0.014 | 0.343±0.036 | 0.322±0.012 | 0.336±0.015 | 0.288±0.018 | 0.269±0.028 |
| | L2/L3* | 0.289±0.027 | 0.366±0.042 | – | – | – | – | – | – |
| | L3/L4* | 0.312±0.009 | 0.350±0.031 | – | – | – | – | – | – |
| | L4/L5 | 0.303±0.036 | 0.334±0.036 | 0.363±0.021 | 0.300±0.051 | 0.335±0.023 | 0.319±0.008 | 0.299±0.009 | 0.247±0.018 |

Disc height index values were calculated using the formula: (average of mid-sagittal craniocaudal disc height values measured using the software's line tool at dorsal, middle, and ventral disc locations)/craniocaudal length of the vertebral body cranial to the disc, measured at the middle location. Bone to bone measurements were based on placement of the electronic cursors on osseous vertebral endplate margins adjacent to the disc. Blank cells indicate locations where the observer could not sufficiently visualize margins for measurements. Asterisks indicate surgically injured disc locations.

**Table 5. Average of triplicate DHI Index values recorded by two observers using the endplate-to-endplate measurement method for each goat, time point, and intervertebral disc level.**

| Goat ID# | Disc Level | Disc Height Index Measured Endplate-Endplate (Mean±SEM) | | | | | | | |
|---|---|---|---|---|---|---|---|---|---|
| | | Baseline | | 3 Week | | 6 Week | | 12 Week | |
| | | Observer 1 | Observer 2 | Observer 1 | Observer 2 | Observer 1 | Observer 2 | Observer 1 | Observer 2 |
| 77 | L1/L2* | 0.178±0.004 | 0.165±0.021 | – | – | – | – | – | – |
| | L2/L3* | 0.161±0.009 | 0.145±0.024 | – | – | – | – | – | – |
| | L3/L4 | 0.165±0.004 | 0.140±0.020 | 0.199±0.050 | 0.146±0.014 | 0.199±0.010 | 0.180±0.001 | 0.184±0.005 | 0.164±0.009 |
| | L4/L5 | 0.168±0.011 | 0.144±0.032 | – | – | 0.157±0.008 | 0.152±0.021 | 0.172±0.016 | 0.143±0.007 |
| 79 | L1/L2* | 0.204±0.015 | 0.176±0.025 | 0.198±0.012 | 0.170±0.013 | 0.182±0.019 | 0.176±0.017 | 0.190±0.006 | 0.193±0.013 |
| | L2/L3* | 0.200±0.020 | 0.177±0.034 | 0.184±0.012 | 0.155±0.017 | 0.183±0.019 | 0.154±0.009 | 0.177±0.003 | 0.167±0.008 |
| | L3/L4 | 0.213±0.027 | 0.166±0.045 | 0.187±0.006 | 0.157±0.024 | 0.188±0.025 | 0.155±0.009 | 0.198±0.006 | 0.163±0.012 |
| | L4/L5 | 0.180±0.028 | 0.167±0.039 | 0.182±0.002 | 0.166±0.016 | 0.203±0.023 | – | 0.185±0.006 | 0.171±0.007 |
| 80 | L1/L2 | 0.180±0.013 | 0.187±0.021 | 0.204±0.013 | 0.181±0.018 | 0.174±0.009 | 0.175±0.006 | 0.178±0.017 | 0.159±0.014 |
| | L2/L3* | 0.164±0.002 | 0.173±0.023 | – | – | – | – | – | – |
| | L3/L4* | 0.173±0.008 | 0.169±0.024 | – | – | – | – | – | – |
| | L4/L5 | 0.248±0.011 | 0.151±0.022 | 0.225±0.017 | 0.149±0.016 | 0.206±0.014 | 0.167±0.014 | 0.177±0.002 | 0.157±0.011 |

Disc height index values were calculated using the formula: (average of mid-sagittal craniocaudal disc height values measured using the software's line tool at dorsal, middle, and ventral disc locations)/craniocaudal length of the vertebral body cranial to the disc, measured at the middle location. Endplate to endplate measurements were based on placement of electronic cursors on cartilaginous vertebral endplate margins adjacent to the disc. Blank cells indicate locations where the observer could not sufficiently visualize margins for cursor placement. Asterisks indicate surgically injured disc locations.

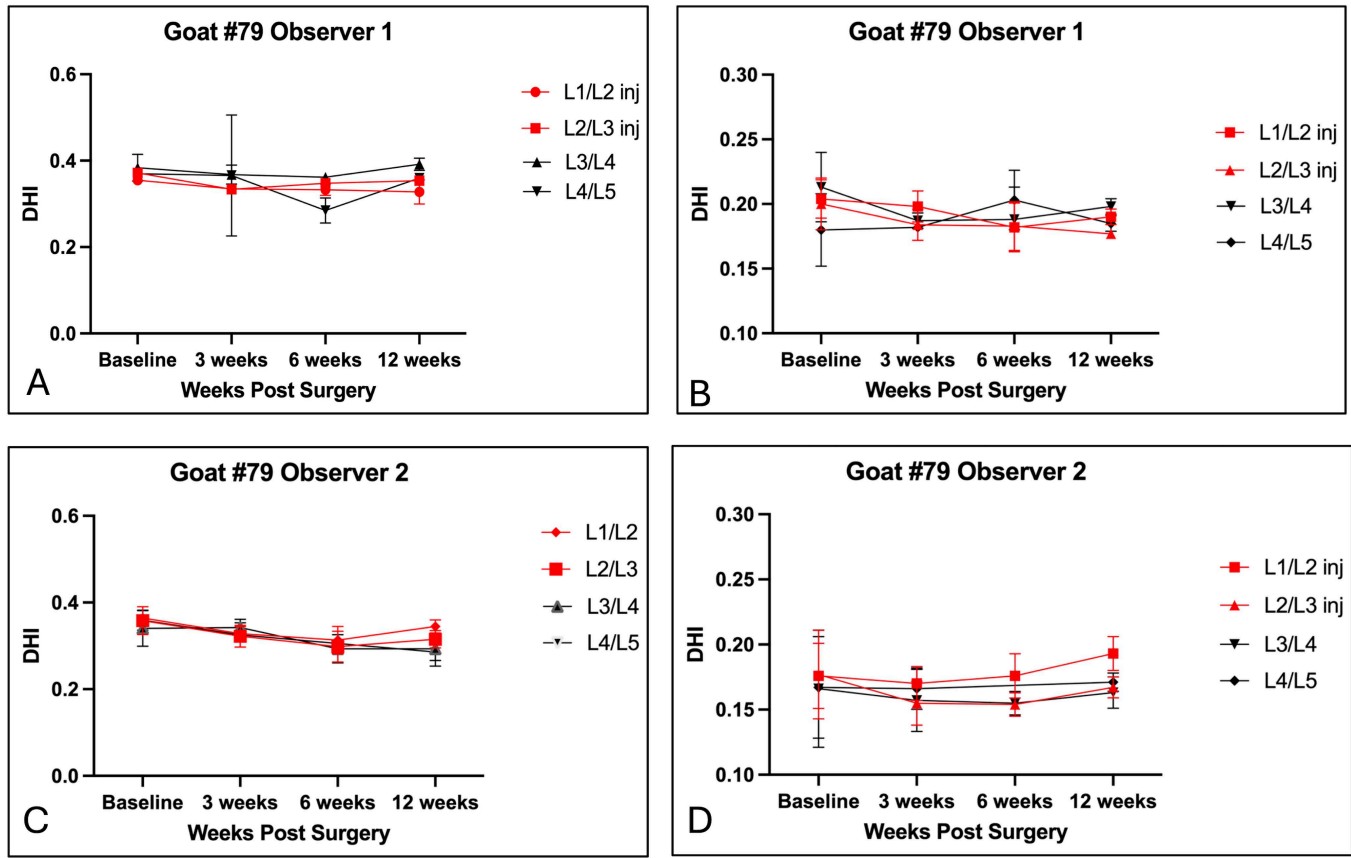

**Fig 5. Graph illustrating triplicate measures of disc height index (DHI) for Goat # 79, recorded by two observers using adjacent vertebral bone margins (bone-to-bone) and using adjacent vertebral cartilaginous margins (endplate-to-endplate) at each of the time points.** (Goats 77 and 80 were excluded from analyses due to inflammatory responses and loss of visualization of vertebral margins.) **A.** Observer 1, bone-to-bone. **B.** Observer 1, endplate-to-endplate. **C.** Observer 2. Bone-to-bone. **D.** Observer 2. Endplate-to-endplate. Bone-to-bone measurements subjectively yielded more consistent DHI measures for both observers.

bone and IVD were found at the locations where microdiscectomy injuries were induced (Fig 8). Dissection of the entire thoracolumbar spine demonstrated hard, proliferative tissue surrounding the injured discs. Evaluation of post-mortem radiographs of the formalin-fixed vertebral specimens for injured disc locations indicated defects in vertebral endplate margins, increased subchondral bone opacity, and proliferative bone surrounding vertebral endplates (spondylosis deformans, ankylosing spondylitis) with varying degrees of severity. Un-injured disc locations did not exhibit these changes. Evaluation of formalin-fixed gross slice images of the specimens indicated either unremarkable IVD and surrounding soft tissues and bones, or severe IVD changes characterized by mild to moderate enlargement with spondylosis deformans (ankylosing spondylitis) and complete effacement of IVDs by the proliferation of firm, pale, fibrous-like connective tissue resulting in complete articular fusion. The mid-vertebral bodies were unremarkable, ruling out osteomyelitis. Detailed descriptions of histological findings are provided in Table 6. Sections for the 4 injured discs from Goat 77 and Goat 80 demonstrated that the nucleus pulposus was replaced by moderately unorganized, fibrous reactive connective tissue, moderately vascularized with minimal inflammatory infiltrates, mostly lymphocytic. In two of these discs, fibrosis was less prominent, with the presence of cartilaginous metaplasia and bone trabeculae, probably more chronic processes leading to ankylosis. No characteristics of acute active inflammation or biological agents (bacteria, fungi)

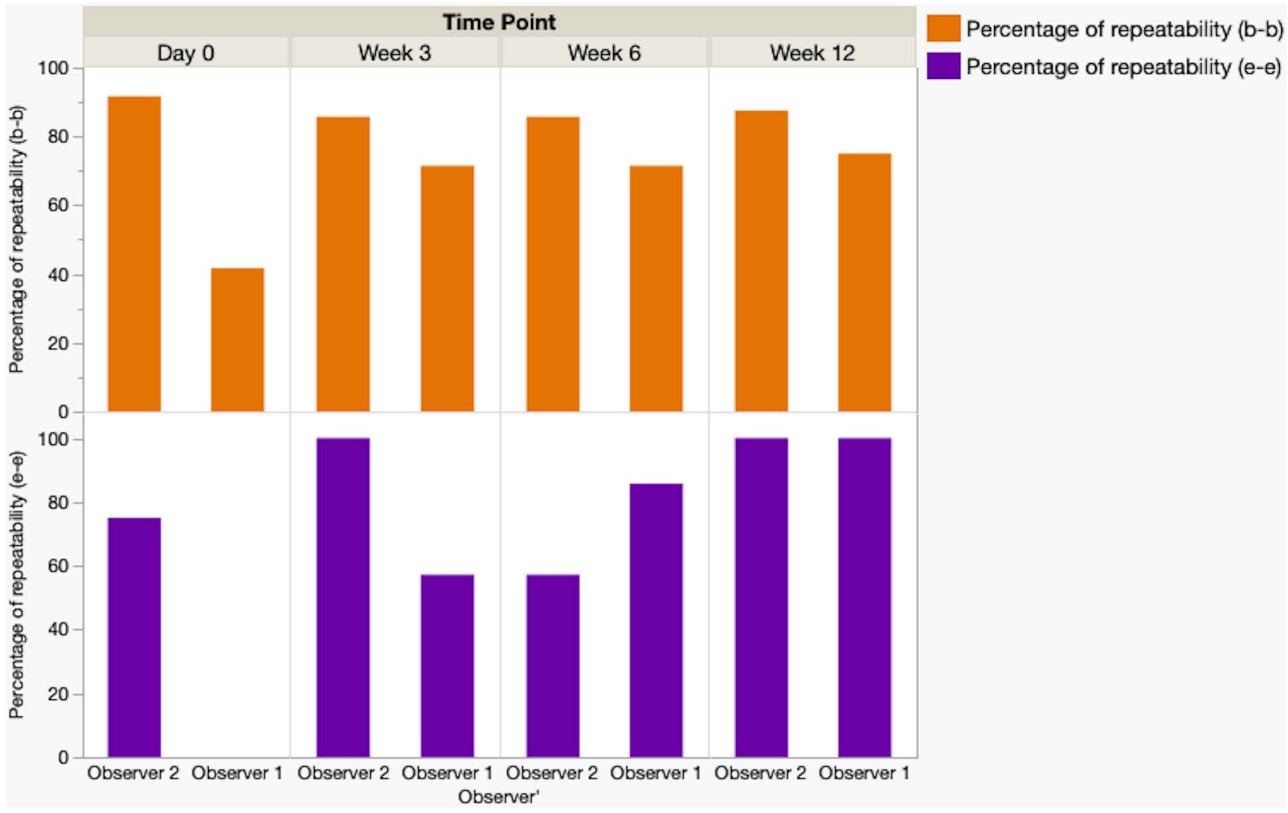

**Fig 6. Graph illustrating percentages of repeatable triplicate measures of disc height index (DHI) by two observers using adjacent vertebral bone margins (bone-to-bone) and versus using adjacent vertebral cartilaginous margins (endplate-to-endplate).** Measures were defined as repeatable if the calculated coefficients of variation (CVs) for the triplicate measures were 10 or below. Observer 2 showed higher repeatability than Observer 1, particularly for bone-to-bone measurements. However, Observer 1 demonstrated a progressive increase in repeatability with the progression of time points.

were seen. Pathologic findings were consistent with chronic, non-infectious (aseptic) discospondylitis. No substantial microscopic lesions were detected in the remaining 8 intervertebral discs examined for Goats 77, 79, or 80.

## Discussion

Our overall aim to determine whether lf-MRI would be a feasible technique for longitudinally assessing surgical microdiscectomy-induced lumbar disc injury and degeneration in goat models was partially achieved. Using pathologic assessment as the reference standard, our standardized lf-MRI image acquisition protocol yielded correct diagnoses of normal IVDs and correct diagnoses of unexpected, non-infectious discospondylitis at injured discs for Goats 77 and 80. Longitudinal lf-MRI in these two goats allowed early detection of acute, progressive characteristics of non-infectious discospondylitis at 3 and 6 weeks, and detection of partial resolution of characteristics at the terminal 12-week timepoint. These findings have not been previously reported. The scanning protocol required anesthesia times ranging from 30 to 60 minutes (median 47.5 minutes) and yielded good quality IVD images with characteristics that mimicked T2 images. The goats tolerated the repeated anesthesia episodes and surgical procedures well and had no clinical signs of distress.

Some unexpected findings precluded our ability to completely achieve the overall aim of the current study. Standard imaging and pathologic characteristics of IVD degeneration were not found in the injured discs and this precluded our ability to assess the feasibility of using lf-MRI to longitudinally quantify IVD degeneration. Non-infectious discospondylitis at

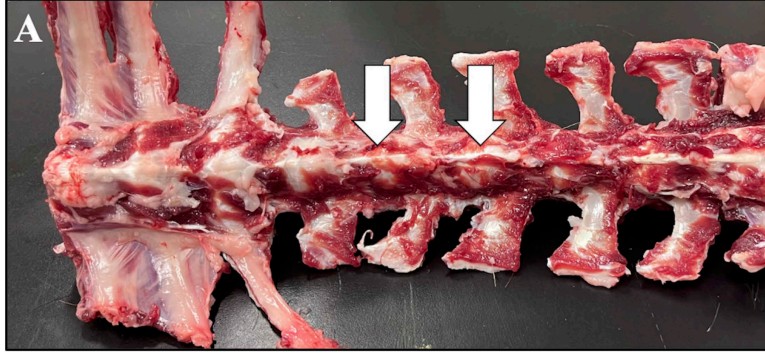
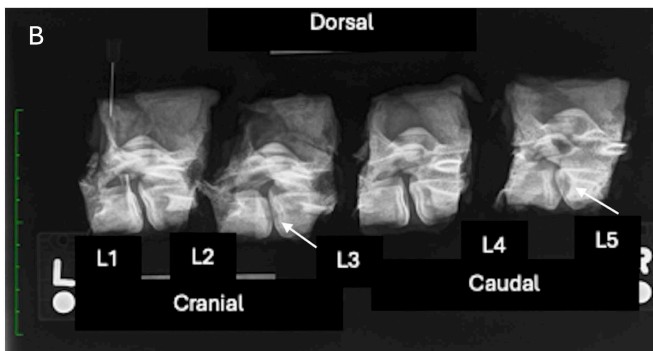
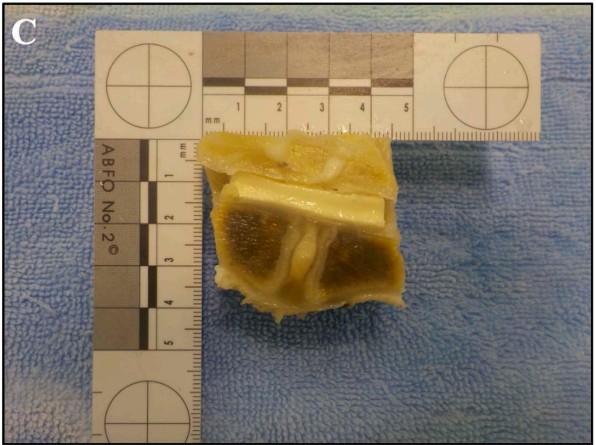
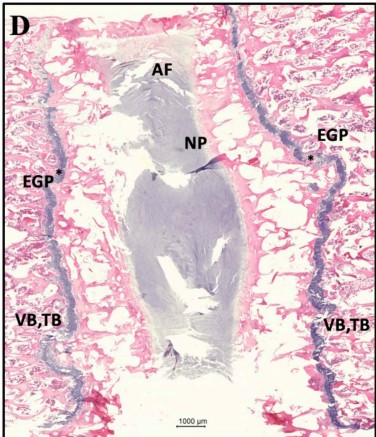
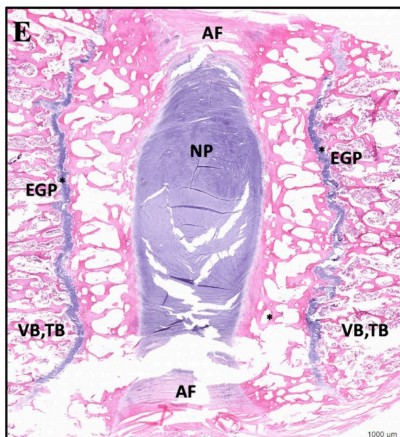

**Fig 7. Images illustrating pathologic characteristics of the least severe IVD injury response in Goat 79.** Gross dissection photograph of the thoracolumbar spine showing no evidence of proliferative tissue formation surrounding the surgically injured IVD levels (white arrows) **(A)**. Radiographs of excised, formalin-fixed vertebral segments demonstrating small endplate defects (small white arrows) **(B)**. Photograph of the formalin-fixed, gross specimen for the surgically injured disc levels following median transection. No evidence of lesion formation is seen in either of the two IVDs **(C)**. Selected representative histopathology H&E images of a surgically injured disc level **(D)** and a comparison non-injured disc level **(E)**. No definitive signs of IVD degeneration or inflammation are evident in either location. Label key = AF – annulus fibrosus, NP – nucleus pulposus, EGP – Endplate Growth Plate marked with *, VB, TB – Vertebral subchondral bone, Trabecular bone.

injured discs for Goats 77 and 80 precluded Pfirrmann or Modic scoring, and measurements of MRI Index or DHI [20]. For the two goats with unexpected non-infectious discospondylitis at both of their injured disc locations, measurements such as Modic scoring, Pfirrmann grading, MRI Index, and Disc Height Index were not possible. The imaging findings mimicked MRI characteristics of infectious discospondylitis in animals or spondylodiscitis in humans [21–23]. However, neither of the affected goats exhibited clinical signs of pain, lameness, decreased appetite, or fever and neither of the affected goats had pathologic evidence of active inflammation or infection at the 12-week terminal timepoint for the injured IVDs. No previously published studies were found describing non-infectious discospondylitis in goats, although non-infectious spondylodiscitis has been described as an immune-mediated response in humans [24]. There was no evidence of an immune-mediated response in pathologic assessments for the two affected goats in our study. It is possible that the vertebral endplates could have been injured during the micro-discectomy procedures, and this mechanical trauma could have induced a local inflammatory response that had partially healed by the 12-week timepoint. Previous studies have indicated that nucleus pulposus material can induce local inflammatory responses and fibrous tissue formation when coming into contact with tissues outside the annulus fibrosus [25]. Proposed mechanisms in humans have included release of inflammatory cytokines including interleukin and tissue necrosis factor and associated release of prostaglandin E2 from histiocytes [26]. The

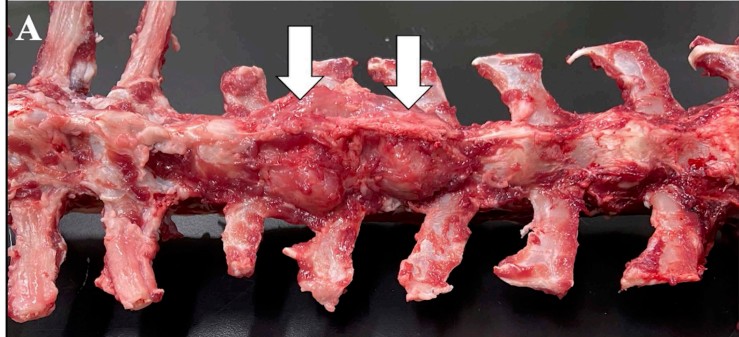
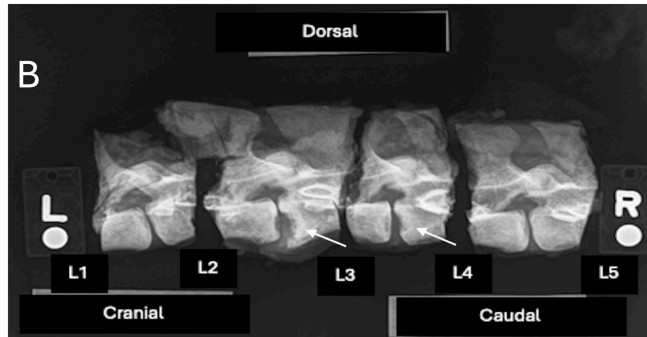
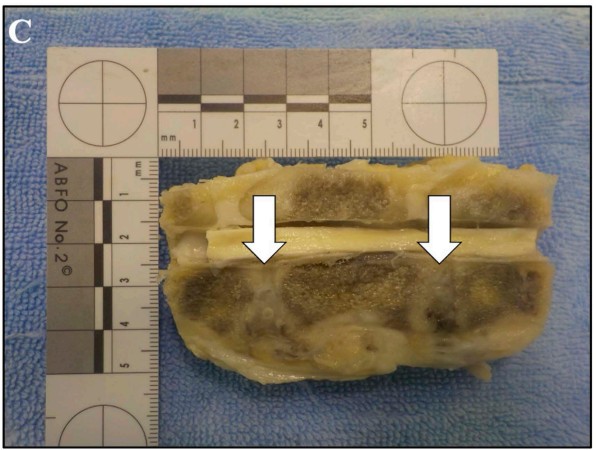
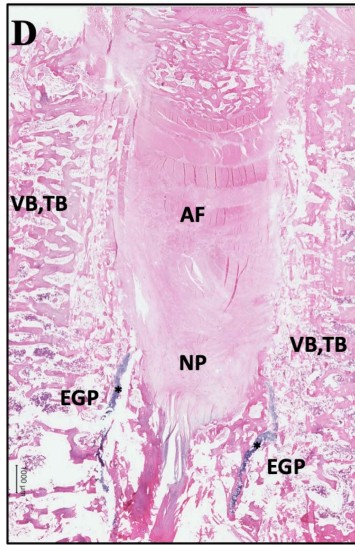
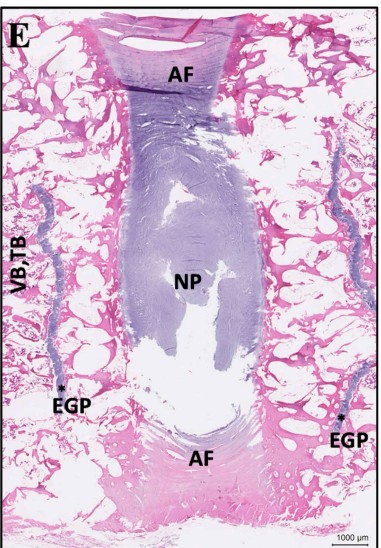

**Fig 8. Images illustrating pathologic characteristics of the most severe IVD injury response in Goat 80.** Gross dissection photograph of the thora-columbar spine demonstrating hard proliferative tissue surrounding the surgically injured IVDs and adjacent vertebrae (white arrows) **(A)**. Radiographs of excised, formalin-fixed vertebral segments illustrating defects in vertebral endplate margins, increased subchondral bone opacity, and proliferative bone surrounding vertebral endplates at the injured disc levels (small white arrows) **(B)**. Photograph of the formalin-fixed, gross specimen for the surgically injured disc levels following median transection. Lesions within both intervertebral discs and their adjacent vertebral endplates are evident (white arrows) **(C)**. Selected representative histopathology H&E images of a surgically injured disc level **(D)** and a comparison non-injured level **(E)**. The surgically injured disc has been replaced by moderately unorganized, fibrous reactive connective tissue and the vertebral bone adjacent to the disc appears moderately vascularized with minimal inflammatory infiltrates, mostly lymphocytic. The vertebral endplate growth plates appear disrupted. Label key: AF – annulus fibrosus, NP – nucleus pulposus, EGP – Endplate Growth Plate marked by *, VB, TB – Vertebral subchondral bone, Trabecular bone.

possibility of instrumentation-induced contamination cannot completely be excluded, however the exact same surgical procedures were used for all goats and one of the goats had no evidence of discospondylitis. More definitive assessment of the non-infectious discospondylitis in affected goats would have required additional lf-MRI sequences. For human patients with suspected infectious or inflammatory spinal disease, additional fat-suppressed T2-weighted, STIR, and fat-suppressed T1-weighted pulse sequences without and with gadolinium contrast are recommended [27,28].

As with any study design, limitations exist. The first limitation of this work was that a small sample size was utilized. However, our study was designed to be an exploratory pilot in order to 1) assess feasibility, 2) identify technical challenges, 3) optimize surgical technique and imaging sequences and, 4) design statistically powered future experiments. The second limitation was a lack of contrast enhanced MRI images. We did not add these sequences for the affected goats in our study because they were asymptomatic and the risks for extending the anesthetic sessions were not

**Table 6. Descriptions of histopathologic characteristics for each goat, based on a consensus of three observers.**

**Goat # 77**

- Lesions were noted in both injured levels
  ◦ Opinion: Overall disc collapse seemed less severe compared to Goat 80
- Histological H&E staining showed the injured levels had a loss of disc integrity
- Histological H&E showed the uninjured levels had no signs of IVD degeneration.

Goat # 79

- No lesions were noted in either the injured or uninjured levels
- Histological H&E staining showed no presence of IVD degeneration in either the injured or uninjured levels

Goat # 80

- Lesions were noted in both injured levels
- Lesions were not noted in both uninjured levels
- Histological H&E staining showed the injured levels had a loss of disc integrity and potential osteophyte formation.
- Histological H&E staining showed the uninjured levels were largely intact and no signs of IVD degeneration.
- Histological processing artifacts were noted

Legend: One observer was an ABVP-certified veterinary pathologist and two observers were biomedical engineers with a research focus in intervertebral disc injury.

considered to be justified. Prolonged anesthetic procedures in goats can increase their risks for complications such as aspiration pneumonia and gut atony [28]. A third limitation was observer variability for some quantitative measurements. Possible reasons could have been communication breakdowns between observers regarding which discs to measure, digital eye strain [20], insufficient Observer 2 experience with MRI interpretations, or decreased conspicuity of margins for discs on the cranial and caudal edges of the scan fields of view [20]. For future studies, it may be helpful to add more training sessions with an experienced observer and to clarify instructions in the image analysis protocol regarding which discs to measure and the importance of taking frequent breaks. It may also be helpful to add more instructions and example images on how to make decisions when there are variations in conspicuity of IVD or endplate margins. Other possible protocol improvements could include the use of automated image analyses or dual readings.

## Conclusions

The current study introduced standardized lf-MRI image acquisition and image analysis protocols for qualitative and quantitative lumbar IVD assessments in goats and novel lf-MRI characteristics of non-infectious discospondylitis at surgically injured disc locations in two goats. These findings can be used as background for future studies evaluating the feasibility of using lf-MRI as a technique for longitudinally measuring IVD injury and degeneration in goat models.

## Supporting information

**S1 Appendix. Standardized Protocol for Goat Lumbar IVD lf-MRI Scanning.**
(DOCX)

**S2 Appendix. Standardized Protocol for Goat Lumbar IVD lf-MRI Measurements.**
(DOCX)

**S3 Appendix. Lf-MRI findings recorded by a veterinary radiologist immediately following image acquisition for each goat at each time point.**
(DOCX)

**S4 Appendix. Disc Height Index (DHI) repeatability analysis details.**
(DOCX)

**S5 Appendix. Disc Height Index (DHI) and Co-efficients of variation (CVs) for each observer, time point, animal ID, disc level, and measurement method.** Measurements were recorded in triplicate for all discs unless they were excluded from analyses due to non-infectious discospondylitis making margins of the nucleus pulposus and vertebral endplates indistinguishable.
(DOCX)

**S6 Appendix. Spreadsheets detailing measurements recorded by each of the two observers.**
(XLSX)

## Acknowledgments

Authors would like to acknowledge the staff of the Clemson Godley-Snell Research Center for their assistance with goat husbandry and acquiring MRI data.

## Author contributions

**Conceptualization:** Jeryl C. Jones, Mario J. Krussig, Matthew W. Breed, Cerano D. Harrison, John W. Gilpin, Guillermo M. Rimoldi, Jeremy J. Mercuri, Ahmed A. B. Ali.

**Data curation:** Jeryl C. Jones, Mario J. Krussig, Matthew W. Breed, Cerano D. Harrison, John W. Gilpin, Guillermo M. Rimoldi, Jeremy J. Mercuri.

**Formal analysis:** Jeryl C. Jones, Mario J. Krussig, Cerano D. Harrison, John W. Gilpin, Guillermo M. Rimoldi, Jeremy J. Mercuri, William C. Bridges.

**Funding acquisition:** Jeryl C. Jones, Jeremy J. Mercuri.

**Investigation:** Jeryl C. Jones, Mario J. Krussig, Matthew W. Breed, Cerano D. Harrison, John W. Gilpin, Guillermo M. Rimoldi, Jeremy J. Mercuri.

**Methodology:** Jeryl C. Jones, Mario J. Krussig, Matthew W. Breed, Cerano D. Harrison, John W. Gilpin, Guillermo M. Rimoldi, Jeremy J. Mercuri, Ahmed A. B. Ali, William C. Bridges.

**Project administration:** Jeryl C. Jones, Matthew W. Breed, Jeremy J. Mercuri.

**Resources:** Matthew W. Breed, Jeremy J. Mercuri.

**Supervision:** Jeryl C. Jones, Matthew W. Breed, Cerano D. Harrison, Jeremy J. Mercuri, William C. Bridges.

**Validation:** Jeryl C. Jones, Mario J. Krussig, Matthew W. Breed, Cerano D. Harrison, John W. Gilpin, Guillermo M. Rimoldi, Jeremy J. Mercuri, William C. Bridges.

**Visualization:** Jeryl C. Jones, Mario J. Krussig, Cerano D. Harrison, John W. Gilpin, Guillermo M. Rimoldi, Jeremy J. Mercuri.

**Writing – original draft:** Jeryl C. Jones, Mario J. Krussig, Matthew W. Breed, Cerano D. Harrison, John W. Gilpin, Guillermo M. Rimoldi, Ahmed A. B. Ali.

**Writing – review & editing:** Jeryl C. Jones, Mario J. Krussig, Matthew W. Breed, Cerano D. Harrison, John W. Gilpin, Guillermo M. Rimoldi, Jeremy J. Mercuri, Ahmed A. B. Ali, William C. Bridges.

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
