## [Decision Letter · Decision Letter 0]

29 Aug 2025

Dear Dr. Jones,

Thank you for submitting your manuscript to PLOS ONE. After careful consideration, we feel that it has merit but does not fully meet PLOS ONE’s publication criteria as it currently stands. Therefore, we invite you to submit a revised version of the manuscript that addresses the points raised during the review process.

Please carefully implement all recommendations and suggestions in the manuscript.

We look forward to receiving your revised manuscript.

Kind regards,

Svenja Illien-Jünger, Ph.D.

Academic Editor

PLOS ONE

Journal Requirements:

2. Please expand the acronyms “NIH NIGMS” and “NIFA/USDA” (as indicated in your financial disclosure) so that it states the name of your funders in full.

Funding support was provided by the South Carolina Center of Biomedical Research Excellence for Translational Research Improving Musculoskeletal Health (SC TRIMH, NIH NIGMS P20 GM121342) and the South Carolina Bioengineering Center for Regeneration and Formation of Tissues (SC BioCRAFT, NIH NIGMS).  This material is also based upon work supported by the NIFA/USDA, under Multistate Project number SC- 1700608 (NC-1029). Technical Contribution No. 7433 of the Clemson University Experiment Station.

5. Please amend your list of authors on the manuscript to ensure that each author is linked to an affiliation. Authors’ affiliations should reflect the institution where the work was done (if authors moved subsequently, you can also list the new affiliation stating “current affiliation:….” as necessary).

6. Please include your tables as part of your main manuscript and remove the individual files. Please note that supplementary tables (should remain/ be uploaded) as separate "supporting information" files

Reviewers' comments:

Reviewer's Responses to Questions

**Comments to the Author**

1. Is the manuscript technically sound, and do the data support the conclusions?

Reviewer #1: Partly

Reviewer #2: Partly

2. Has the statistical analysis been performed appropriately and rigorously?

Reviewer #1: No

Reviewer #2: No

3. Have the authors made all data underlying the findings in their manuscript fully available?

Reviewer #1: Yes

Reviewer #2: No

4. Is the manuscript presented in an intelligible fashion and written in standard English?

Reviewer #1: Yes

Reviewer #2: Yes

Reviewer #1: The paper is thorough in its experimental design, ethics, and data presentation. However, a few revisions are necessary to improve clarity, interpretation, and overall scientific communication.

1. I recommend further emphasizing how LF-MRI may serve as a viable alternative to high-field MRI, specifically in preclinical studies. Including a sentence or two comparing the diagnostic accuracy and clinical translatability of the two systems could enhance context.

2. Could mechanical trauma during microdiscectomy or instrumentation-induced contamination be potential causes, even in the absence of infection? Please discuss possible pathophysiological mechanisms more explicitly, ideally supported by citations.

3. The inter- and intra-observer variability for quantitative measurements (DHI and MRI index) was high. The authors mention digital eye strain and observer inexperience. These are valid points, but please consider suggesting how such variability might be addressed in future studies (e.g., automated image analysis, training sessions, dual readings).

4. The authors primarily use descriptive statistics and coefficients of variation. This is acceptable for a pilot study; however, adding confidence intervals where appropriate (e.g., for CVs or MRI index) would improve statistical rigor.

5. There are sections with overly long sentences. For example, in the introduction and results, several sentences exceed 40 words, making them difficult to follow. Consider splitting complex sentences to enhance readability.

6. he terms "non-infectious discospondylitis" and "spondylodiscitis" are used interchangeably. While both are technically correct, the authors should clearly define their preferred terminology and use it consistently.

7. The figure legends could be improved for clarity (e.g., clearly marking the specific discs affected and providing a scale or orientation bar). Could you make sure the image resolution is suitable for publication?

8. Although some limitations are acknowledged, please consider explicitly listing them in the discussion as a separate paragraph for clarity. For instance:

Small sample size

Lack of contrast-enhanced sequences

Observer variation

Limited histopathologic sampling

Reviewer #2: The authors successfully developed standardized imaging and surgical protocols and demonstrated that lf-MRI can produce diagnostically useful images with acceptable anesthesia times and minimal animal distress. The identification of previously unreported lf-MRI features of non-infectious discospondylitis in goats is a noteworthy contribution to translational imaging research. However, I have several concerns regarding the statistical methodology used to assess measurement reliability:

1. Reproducibility of Reported Results

As I am a statistician, I attempted to reproduce the analyses described in Appendix 4 using the data provided in Appendix 5. I was able to replicate the reported mean values of the coefficients of variation (CVs) and the proportions of CV measures classified as not repeatable. However, I was unable to reproduce the p-values reported by the authors.

The manuscript does not specify whether the t-tests used to compare CVs between observers were paired or unpaired. To investigate this, I conducted both paired and unpaired Student’s t-tests comparing the mean CVs between the two observers. In neither case did I obtain the same p-values as those reported.

2. Suitability of Parametric Tests

The use of Student’s t-tests is questionable given the small sample sizes involved. Parametric tests such as the t-test rely on assumptions (e.g., normality, homogeneity of variance) that may not be met under these conditions. For example, at weeks 3, 6, and 12, several discs were excluded from analysis due to inflammatory responses and poor image quality, further reducing the number of valid observations and weakening statistical power.

Under such conditions, a more appropriate approach would have been the use of non-parametric tests. Given that measurements were performed on the same discs by both observers, a paired t-test would have been the statistically appropriate choice. So, I recommend the Wilcoxon signed-rank test for paired data. This method is better suited for small samples and do not assume normality, providing more robust and reliable inference.

3. Limitations of CV-Based Analysis

The authors compared CVs between observers to evaluate the repeatability of MRI Index and Disc Height Index (DHI) measurements. While CV analysis is useful for assessing intra-observer precision, it does not capture inter-observer agreement. CVs quantify relative variability within repeated measurements by a single observer but do not indicate whether different observers produce concordant results when evaluating the same anatomical structures.

Given the study’s emphasis on validating lf-MRI as a reliable tool for longitudinal assessment, a more appropriate statistical approach would have been the use of Intraclass Correlation Coefficients (ICC). ICC provides a direct measure of inter-observer reliability by estimating the proportion of total variance attributable to true differences between subjects versus measurement error. Its use is standard in imaging studies where reproducibility across raters is critical. The absence of ICC analysis, or complementary methods such as Bland-Altman plots, limits the interpretability of the study’s findings regarding measurement robustness.

4. Data Transparency

Given the small sample size, I recommend that the authors provide the raw MRI Index and DHI measurements collected by both observers in a Supplemental Appendix. Making the individual data points available would allow for independent verification of the statistical analyses and enhance the transparency and reproducibility of the study.

Conclusion

While the study offers valuable insights into the technical feasibility of lf-MRI in goat models and introduces novel imaging findings, the statistical framework used to evaluate measurement reliability falls short of current best practices. I recommend that the authors:

- Clarify the statistical methods used, particularly the type of t-tests applied.

- Reanalyze the data using appropriate non-parametric techniques.

- Incorporate ICC and complementary methods (Bland-Altman plot) to assess inter-observer reproducibility.

- Provide raw measurement all data to support transparency and reproducibility.

These revisions would significantly strengthen the methodological rigor and reliability of the study’s conclusions.

**Do you want your identity to be public for this peer review?** For information about this choice, including consent withdrawal, please see our Privacy Policy

Reviewer #1: No

Reviewer #2: **Yes:** KEILA MARA CASSIANO

---

## [Author Response · Author response to Decision Letter 1]

3 Nov 2025

Editor Comments Author Response

Thank you, we have revised the style as requested.

2. Please expand the acronyms “NIH NIGMS” and “NIFA/USDA” (as indicated in your financial disclosure) so that it states the name of your funders in full.

Thank you, definitions for these abbreviations have been provided in the cover letter.

Thank you, the SC BioCRAFT grant had partially supported our study but that grant has now ended. We are unsure how you want us to indicate this and would appreciate advice.

Funding support was provided by the South Carolina Center of Biomedical Research Excellence for Translational Research Improving Musculoskeletal Health (SC TRIMH, NIH NIGMS P20 GM121342) and the South Carolina Bioengineering Center for Regeneration and Formation of Tissues (SC BioCRAFT, NIH NIGMS). This material is also based upon work supported by the NIFA/USDA, under Multistate Project number SC- 1700608 (NC-1029). Technical Contribution No. 7433 of the Clemson University Experiment Station.

Thank you, funders had none of the listed roles and we have added this information to the cover letter.

5. Please amend your list of authors on the manuscript to ensure that each author is linked to an affiliation. Authors’ affiliations should reflect the institution where the work was done (if authors moved subsequently, you can also list the new affiliation stating “current affiliation:….” as necessary).

Thank you, the author list and affiliations have been revised as requested. We have added a statistician as a new author and provided his affiliation.

6. Please include your tables as part of your main manuscript and remove the individual files. Please note that supplementary tables (should remain/ be uploaded) as separate "supporting information" files

Thank you, we have inserted tables into the main document and removed the previously uploaded individual files.

Thank you. We did not see reviewer requests for specific citations.

5. Review Comments to the Author

Reviewer #1: The paper is thorough in its experimental design, ethics, and data presentation. However, a few revisions are necessary to improve clarity, interpretation, and overall scientific communication.

Thank you for your efforts to help us improve our paper.

1. I recommend further emphasizing how LF-MRI may serve as a viable alternative to high-field MRI, specifically in preclinical studies. Including a sentence or two comparing the diagnostic accuracy and clinical translatability of the two systems could enhance context.

Thank you, we added references and descriptions of previous studies comparing diagnostic capability of lf-MRI versus hf-MRI for characterizing IVDs in humans and dogs.

2. Could mechanical trauma during microdiscectomy or instrumentation-induced contamination be potential causes, even in the absence of infection? Please discuss possible pathophysiological mechanisms more explicitly, ideally supported by citations.

Thank you, we have added these possible causes to the Discussion section.

3. The inter- and intra-observer variability for quantitative measurements (DHI and MRI index) was high. The authors mention digital eye strain and observer inexperience. These are valid points, but please consider suggesting how such variability might be addressed in future studies (e.g., automated image analysis, training sessions, dual readings).

Thank you, we have added these examples in the Discussion section.

4. The authors primarily use descriptive statistics and coefficients of variation. This is acceptable for a pilot study; however, adding confidence intervals where appropriate (e.g., for CVs or MRI index) would improve statistical rigor.

Thank you, CV values have been added to the Results.

5. There are sections with overly long sentences. For example, in the introduction and results, several sentences exceed 40 words, making them difficult to follow. Consider splitting complex sentences to enhance readability.

Thank you, we have split long sentences where we could so without altering the intended scientific meaning.

6. The terms "non-infectious discospondylitis" and "spondylodiscitis" are used interchangeably. While both are technically correct, the authors should clearly define their preferred terminology and use it consistently.

Thank you, we have used the term “non-infectious discospondylitis” throughout.

7. The figure legends could be improved for clarity (e.g., clearly marking the specific discs affected and providing a scale or orientation bar). Could you make sure the image resolution is suitable for publication?

We have added more labels to the pathologic assessment figures and explained these in the captions. We also ensured the resolution of each figure is 300-600 dpi.

8. Although some limitations are acknowledged, please consider explicitly listing them in the discussion as a separate paragraph for clarity. For instance:

Small sample size

Lack of contrast-enhanced sequences

Observer variation

Limited histopathologic sampling

Thank you, we have added a limitations paragraph to the Discussion, moved narrative related to these limitations here, and expanded on why these decisions were made. We were unsure what the reviewer meant by “limited histopathologic sampling”. We added language to the Methods clarifying that all lumbar disc locations for each of the goats were examined. The figures provided were intended to be selected examples to illustrate key points.

Reviewer #2: The authors successfully developed standardized imaging and surgical protocols and demonstrated that lf-MRI can produce diagnostically useful images with acceptable anesthesia times and minimal animal distress. The identification of previously unreported lf-MRI features of non-infectious discospondylitis in goats is a noteworthy contribution to translational imaging research. However, I have several concerns regarding the statistical methodology used to assess measurement reliability:

Thank you for your efforts to help us improve our paper. We have added a statistician as a co-author and hope the responses below and revised manuscript documents satisfactorily address your concerns.

1. Reproducibility of Reported Results

As I am a statistician, I attempted to reproduce the analyses described in Appendix 4 using the data provided in Appendix 5. I was able to replicate the reported mean values of the coefficients of variation (CVs) and the proportions of CV measures classified as not repeatable. However, I was unable to reproduce the p-values reported by the authors.

The manuscript does not specify whether the t-tests used to compare CVs between observers were paired or unpaired. To investigate this, I conducted both paired and unpaired Student’s t-tests comparing the mean CVs between the two observers. In neither case did I obtain the same p-values as those reported.

Thank you for your feedback. To clarify, we performed paired t-tests based on a statistical model that included observer, animal, and disc to compare the mean CV values calculated for the measurements of DHI and MRI index (disc hydration), performed in triplicate by the two observers, to assess interobserver repeatability. The analyses were performed using JMP statistical software. Further consultation with a statistician with over 30 years of experience (WB) yielded the same results when the models were re-run. We have provided the original data sets used for the analyses as a new supplemental appendix.

2. Suitability of Parametric Tests

The use of Student’s t-tests is questionable given the small sample sizes involved. Parametric tests such as the t-test rely on assumptions (e.g., normality, homogeneity of variance) that may not be met under these conditions. For example, at weeks 3, 6, and 12, several discs were excluded from analysis due to inflammatory responses and poor image quality, further reducing the number of valid observations and weakening statistical power.

Under such conditions, a more appropriate approach would have been the use of non-parametric tests. Given that measurements were performed on the same discs by both observers, a paired t-test would have been the statistically appropriate choice. So, I recommend the Wilcoxon signed-rank test for paired data. This method is better suited for small samples and do not assume normality, providing more robust and reliable inference.

Thank you, our statistician carefully considered the recommendation to use the Wilcoxon Signed-Rank test and provided the following response: “The Wilcoxon signed-rank test doesn’t assume normality, but has reduced power (in small samples) for detecting differences. We wanted to be sure we found any evidence of differences, and possible inconsistencies, so we chose the regular t-test.”

3. Limitations of CV-Based Analysis

The authors compared CVs between observers to evaluate the repeatability of MRI Index and Disc Height Index (DHI) measurements. While CV analysis is useful for assessing intra-observer precision, it does not capture inter-observer agreement. CVs quantify relative variability within repeated measurements by a single observer but do not indicate whether different observers produce concordant results when evaluating the same anatomical structures.

Given the study’s emphasis on validating lf-MRI as a reliable tool for longitudinal assessment, a more appropriate statistical approach would have been the use of Intraclass Correlation Coefficients (ICC). ICC provides a direct measure of inter-observer reliability by estimating the proportion of total variance attributable to true differences between subjects versus measurement error. Its use is standard in imaging studies where reproducibility across raters is critical. The absence of ICC analysis, or complementary methods such as Bland-Altman plots, limits the interpretability of the study’s findings regarding measurement robustness.

Thank you, we have consulted with our statistician and added an ICC analysis as requested.

4. Data Transparency

Given the small sample size, I recommend that the authors provide the raw MRI Index and DHI measurements collected by both observers in a Supplemental Appendix. Making the individual data points available would allow for independent verification of the statistical analyses and enhance the transparency and reproducibility of the study.

Thank you, we have added a spreadsheet containing all observer measurements as a new supplemental file.

Conclusion

While the study offers valuable insights into the technical feasibility of lf-MRI in goat models and introduces novel imaging findings, the statistical framework used to evaluate measurement reliability falls short of current best practices. I recommend that the authors:

- Clarify the statistical methods used, particularly the type of t-tests applied.

- Reanalyze the data using appropriate non-parametric techniques.

- Incorporate ICC and complementary methods (Bland-Altman plot) to assess inter-observer reproducibility.

- Provide raw measurement all data to support transparency and reproducibility.

These revisions would significantly strengthen the methodological rigor and reliability of the study’s conclusions.

Thank you, we hope the revisions satisfactorily address these concerns.

---

## [Decision Letter · Decision Letter 1]

3 Dec 2025

Dear Dr. Jones,

Thank you for submitting your manuscript to PLOS ONE. After careful consideration, we feel that it has merit but does not fully meet PLOS ONE’s publication criteria as it currently stands. Therefore, we invite you to submit a revised version of the manuscript that addresses the points raised during the review process.

We look forward to receiving your revised manuscript.

Kind regards,

Svenja Illien-Jünger, Ph.D.

Academic Editor

PLOS One

Journal Requirements:

Additional Editor Comments:

Dear Dr. Jones,

thank you for submitting the revised manuscript. While it improved significantly after implementing the reviewers' comments, some concerns regarding the statistic remained. Please implement the concerns that were raised by reviewer 2.

Reviewers' comments:

Reviewer's Responses to Questions

**Comments to the Author**

Reviewer #1: All comments have been addressed

Reviewer #2: All comments have been addressed

2. Is the manuscript technically sound, and do the data support the conclusions?

Reviewer #1: Yes

Reviewer #2: Yes

3. Has the statistical analysis been performed appropriately and rigorously?

Reviewer #1: Yes

Reviewer #2: No

4. Have the authors made all data underlying the findings in their manuscript fully available?

Reviewer #1: Yes

Reviewer #2: Yes

5. Is the manuscript presented in an intelligible fashion and written in standard English?

Reviewer #1: Yes

Reviewer #2: Yes

Reviewer #1: The paper has been improved significantly. I just wanted to let you know that I don't have any additional comments. I think the paper can be accepted for publication in its current form.

Reviewer #2: 1) Please adjust the vertical lines position in Table 2 to improve the separation between columns, as the current format in first columns is confusing.

2) Thank you for the explanation about t-test. However, I would like to maintain the recommendation to replace the paired t-test with the Wilcoxon signed-rank test, for the following reasons:

a) Violation of the normality assumption

The paired t-test requires that the distribution of the paired differences be approximately normal, especially when the sample size is small.

When this assumption is not met, the t-test loses validity and may inflate the Type I error rate or reduce statistical power in unpredictable ways. In other words, the t-test is not robust under non-normality with small samples.

b) Your argument about Wilcoxon having “lower power” needs context…

Although the Wilcoxon signed-rank test may have slightly lower power under perfect normality, it is actually more powerful and more reliable when normality is violated, which is the case here.

Therefore, for the actual data Wilcoxon is better than paired t-test in terms of validity, and the “power” of the t-test is not guaranteed when its assumptions are violated.

c) Methodological rigor requires choosing the test that matches the data distribution

In scientific analysis, the appropriate test is not chosen based on theoretical power alone, but based on whether the assumptions are reasonably satisfied.

Using a parametric test despite known assumption violations compromises: statistical validity, reproducibility, and the credibility of the findings.

Thus, applying the standard paired t-test is not appropriate given the clear violation of assumptions.

3. I find the findings presented in Table 3 rather unusual:

a) The intra-observer ICC for DHI (endplate-to-endplate) and the intra-observer ICC for DHI (bone-to-bone) are identical at booth baseline and week 3, to both observers.

b) The inter-observer ICC for DHI (endplate-to-endplate) is also identical to the inter-observer ICC for DHI (bone-to-bone) at all time points.

These results are highly improbable. Therefore, I recommend carefully revisiting and fully verifying the ICC analysis.

**Do you want your identity to be public for this peer review?** For information about this choice, including consent withdrawal, please see our Privacy Policy

Reviewer #1: No

Reviewer #2: No

---

## [Author Response · Author response to Decision Letter 2]

29 Dec 2025

Reviewer #2: 1) Please adjust the vertical lines position in Table 2 to improve the separation between columns, as the current format in first columns is confusing.

The vertical lines have been adjusted to improve separation between columns containing values

2) Thank you for the explanation about t-test. However, I would like to maintain the recommendation to replace the paired t-test with the Wilcoxon signed-rank test, for the following reasons:

a) Violation of the normality assumption

The paired t-test requires that the distribution of the paired differences be approximately normal, especially when the sample size is small.

When this assumption is not met, the t-test loses validity and may inflate the Type I error rate or reduce statistical power in unpredictable ways. In other words, the t-test is not robust under non-normality with small samples.

b) Your argument about Wilcoxon having “lower power” needs context…

Although the Wilcoxon signed-rank test may have slightly lower power under perfect normality, it is actually more powerful and more reliable when normality is violated, which is the case here.

Therefore, for the actual data Wilcoxon is better than paired t-test in terms of validity, and the “power” of the t-test is not guaranteed when its assumptions are violated.

c) Methodological rigor requires choosing the test that matches the data distribution

In scientific analysis, the appropriate test is not chosen based on theoretical power alone, but based on whether the assumptions are reasonably satisfied.

Using a parametric test despite known assumption violations compromises: statistical validity, reproducibility, and the credibility of the findings.

Thus, applying the standard paired t-test is not appropriate given the clear violation of assumptions.

Thank you for your thorough suggestions and explanations. To address concerns with the chosen methods, we have performed additional analyses for the models run in this study to assess the adherence of the residuals to ANOVA assumptions, including employing rank transformations to demonstrate consistency with previously observed results. The methods section has been expanded for further detail.

3. I find the findings presented in Table 3 rather unusual:

a) The intra-observer ICC for DHI (endplate-to-endplate) and the intra-observer ICC for DHI (bone-to-bone) are identical at booth baseline and week 3, to both observers.

b) The inter-observer ICC for DHI (endplate-to-endplate) is also identical to the inter-observer ICC for DHI (bone-to-bone) at all time points.

These results are highly improbable. Therefore, I recommend carefully revisiting and fully verifying the ICC analysis.

Thank you for bringing this to our attention. The data for bone-to-bone measures were accidentally transposed into the cells for endplate-to-endplate measures. Table 3 now reflects to correct data.

---

## [Decision Letter · Decision Letter 2]

20 Jan 2026

Feasibility of low-field magnetic resonance imaging (lf-MRI) for longitudinally evaluating experimentally induced lumbar intervertebral disc injuries in goat models (Capra hircus): a pilot study

PONE-D-25-26390R2

Dear Dr. Jones,

We’re pleased to inform you that your manuscript has been judged scientifically suitable for publication and will be formally accepted for publication once it meets all outstanding technical requirements.

Kind regards,

Svenja Illien-Jünger, Ph.D.

Academic Editor

PLOS One

Additional Editor Comments (optional):

Reviewers' comments:

Reviewer's Responses to Questions

**Comments to the Author**

Reviewer #2: All comments have been addressed

2. Is the manuscript technically sound, and do the data support the conclusions?

Reviewer #2: Yes

3. Has the statistical analysis been performed appropriately and rigorously?

Reviewer #2: Yes

4. Have the authors made all data underlying the findings in their manuscript fully available?

Reviewer #2: Yes

5. Is the manuscript presented in an intelligible fashion and written in standard English?

Reviewer #2: Yes

Reviewer #2: (No Response)

**Do you want your identity to be public for this peer review?** For information about this choice, including consent withdrawal, please see our Privacy Policy

Reviewer #2: **Yes:** KEILA MARA CASSIANO

---

## [Editor Report · Acceptance letter]

PONE-D-25-26390R2

PLOS One

Dear Dr. Jones,

I'm pleased to inform you that your manuscript has been deemed suitable for publication in PLOS One. Congratulations! Your manuscript is now being handed over to our production team.

Kind regards,

on behalf of

Dr. Svenja Illien-Jünger

Academic Editor

PLOS One